# Resting-State EEG in Alpha Rhythm May Be Indicative of the Performance of Motor Imagery-Based Brain–Computer Interface

**DOI:** 10.3390/e24111556

**Published:** 2022-10-29

**Authors:** Kun Wang, Feifan Tian, Minpeng Xu, Shanshan Zhang, Lichao Xu, Dong Ming

**Affiliations:** 1Tianjin International Joint Research Center for Neural Engineering, Academy of Medical Engineering and Translational Medicine, Tianjin University, Tianjin 300072, China; 2Department of Biomedical Engineering, College of Precision Instruments and Optoelectronics Engineering, Tianjin University, Tianjin 300072, China

**Keywords:** motor imagery, electroencephalogram, resting-state, alpha rhythm, performance variation

## Abstract

Motor imagery-based brain–computer interfaces (MI-BCIs) have great application prospects in motor enhancement and rehabilitation. However, the capacity to control a MI-BCI varies among persons. Predicting the MI ability of a user remains challenging in BCI studies. We first calculated the relative power level (*RPL*), power spectral entropy (*PSE*) and Lempel–Ziv complexity (*LZC*) of the resting-state open and closed-eye EEG of different frequency bands and investigated their correlations with the upper and lower limbs MI performance (left hand, right hand, both hands and feet MI tasks) on as many as 105 subjects. Then, the most significant related features were used to construct a classifier to separate the high MI performance group from the low MI performance group. The results showed that the features of open-eye resting alpha-band EEG had the strongest significant correlations with MI performance. The *PSE* performed the best among all features for the screening of the MI performance, with the classification accuracy of 85.24%. These findings demonstrated that the alpha bands might offer information related to the user’s MI ability, which could be used to explore more effective and general neural markers to screen subjects and design individual MI training strategies.

## 1. Introduction

Brain–computer interface (BCI) is a direct communication pathway between the central nervous system and external devices that replaces, restores, enhances or improves natural central nervous system output [1,2]. Among all BCI paradigms, motor imagery (MI) is one of the most natural BCI paradigms, which is a mental process of motion intention without actual action output. Motor imagery-based BCI (MI-BCI) has been widely applied in clinical rehabilitation, acquisition and the refinement of motor skills [3]. The sensorimotor electroencephalographic (EEG) rhythms could be voluntary modulated by MI, which are, respectively, known as event-related desynchronization/synchronization (ERD/ERS) at mu/alpha (8–13 Hz) and beta (14–30 Hz) bands [4]. Based on this, the neural oscillation patterns induced by different MI tasks could be used for BCI control.

However, the EEG response of the same MI task is varied between subjects, which leads to inter-subjects variability in MI-BCI performance [5,6]. Moreover, in daily experiments, a notable portion of subjects could not use MI to drive BCI successfully [7], which was known as “BCI-illiteracy” or “BCI-inefficiency” [8]. In fact, controlling a MI-BCI requires training to acquire these skills [9]. Thus, the inter-subjects variability in MI-BCI performance has raised questions about how to design the appropriate training strategy to improve users’ ability. In the process of solving the above problem, the prediction of users’ ability is a crucial step. Hence, it is meaningful to find a pre-experimental predictor related to MI performance [10,11]. The predictor of MI performance can not only be used to avoid the loss of time for both users and experimenters, but it can also help us understand the reason for the poor MI performance to guide the design of personalized training protocols [7].

So far, two kinds of predictors have been investigated: psychological and neurophysiological predictors. Daum et al. found that attention might be correlated to the ability of the regulation of slow cortical potentials (SCP) [12]. Burde et al. showed that a person who felt more comfortable with the BCI technology and believed in his/her own ability was likely to yield a good performance [13]. Furthermore, Hammer et al. investigated the correlation between the parameters such as attention span, personality, motivation and so on with the performance of MI-BCIs and found the modulation effect of the psychological parameters on the control performance [14]. In addition, fatigue is directly related to BCI performance. Ferreira et al. showed that MI ability was significantly decreased after intermittent exercise [15]. However, some psychological factors are subjective and are, therefore, not suitable for predicting BCI performance.

At the same time, there are some studies that investigate the neurophysiological correlates of performance variations between subjects. One of the previous studies found that the power of gamma oscillations could modulate the sensorimotor rhythm induced by motor imagery [16]. Ahn et al. analyzed the resting-state EEG data at the eyes-open state of 52 subjects and found that the power of theta band (4~8 Hz) and alpha band (8~13 Hz) were related to MI performance [17]. Zhang et al. discovered that the power spectral entropy (*PSE*) of eyes-closed resting-state EEG in channel C3 was correlated with MI performance through the experiment consisting of 40 subjects [18]. Moreover, Lee et al. demonstrated that the connections of the resting-state network could affect MI performance [11]. Generally speaking, physiological indicators are more objective and convincing than psychological indicators and could be used to guide the training of MI-BCIs, which attract the most researchers in this study topic.

In addition, it should be noted that in the practical application of MI-BCI, the recognition of lower limbs is as important as that of upper limbs [19,20]. However, most of the previous studies are based on the left- and right-hand MI tasks. Therefore, a large public database including both upper and lower limbs MI tasks should be used to find the possible predictors of MI performance. Moreover, researchers often focus on the study of single feature analysis. More sensitive EEG features for MI performance and the joint analysis of multi-dimensional features need further exploration. Hence, this paper studied the correlations between the multi-frequency resting state EEG features and MI performance of four MI tasks (left hand, right hand, both hands and both feet) among more than 100 subjects. Lempel–Ziv complexity (*LZC*) is often used as an EEG feature to detect diseases such as depression. Therefore, to explore resting-state EEG features related to the user’s motor imagery performance, band power features and two non-linear dynamic features (*PSE* and *LZC*) of resting-state EEG were calculated to analyze their correlations with MI performance and construct the prediction models. Finally, other interpretations of the results were discussed.

## 2. Materials and Methods

### 2.1. Database Introduction

All EEG data used in this paper were from EEG Motor Movement/Imagery Dataset [21,22]. A total of 109 subjects participated in the experiment. The experiment included: one one-minute eyes-open resting run; one one-minute eyes-closed resting run; three imagined left/right hand grip runs; and three imagined both hands/feet grip runs. The MI tasks in each run appeared randomly and each task had 25 trials. The EEG data were recorded from 64 electrodes by the BCI2000 system under offline experimental conditions (position of electrodes as shown in Figure 1). The sampling rate was 160 Hz.

### 2.2. Subject Grouping

Due to the abnormal labels of 4 subjects in the database, the data of 105 subjects were analyzed in this paper. The EEG signals of MI tasks were first filtered by an 8–30 Hz Butterworth band-pass filter. Multiclass common spatial pattern (Multi-CSP) was used to extract the features of the EEG patterns induced by four MI tasks, i.e., left/right hand MI and both hands/feet. CSP is one of the most popular spatial filtering methods for the recognition of MI tasks [23]. We selected the eigenvectors corresponding to the two largest eigenvalues for the CSP filter of each class to extract features. A linear support vector machine (SVM) was used to build the multiclass classifier with the help of the famous software package LIBSVM [24]. We selected the default SVM type and set the penalty factor C to 1. The classification accuracies were computed ten times with a ten-fold cross-validation procedure. We utilized the mean classification accuracy to represent the MI performance. The performances of each subject and their standard deviations are shown in Figure 2. The subjects were assigned to three groups: group H (high MI performance, 60~100% accuracy, *N* = 20); group M (medium MI performance, 40~60% accuracy, *N* = 42); and group L (low MI performance, lower than 40% accuracy, *N* = 43). We also calculated the classification accuracies of two classes of MI tasks, i.e., left hand vs. right hand, and both hands vs. feet, to analyze the correlations between EEG features and MI performance.

### 2.3. Resting-State Signal Processing

Eyes-open and -closed resting-state EEG signals were used to analyze the performance variation. Common average reference (CAR) was utilized to increase the signal-to-noise ratio [17]. For the EEG data of each channel, we could record about 9600 data points of a one-minute task. Every 1500 points were taken as an epoch according to the convention. So, we could obtain six epochs of each resting task. For each epoch, relative power level (*RPL*), power spectral entropy (*PSE*) and Lempel–Ziv complexity (*LZC*) in the theta, alpha, beta and gamma bands of eyes-open and eyes-closed resting-state EEG were calculated. Then, the mean of six epochs were the feature of each channel to analyze their correlations with MI performance.

We first used four Butterworth band-pass filters to obtain the theta (4~8 Hz), alpha (8~13 Hz), beta (13~30 Hz) and gamma (30~50 Hz) frequency bands’ EEG signals, respectively. Then, we calculated power of each frequency band as follows:(1)$$P=\frac{1}{N}{\left|x\left(t\right)\right|}^{2}$$

We normalized the band powers by using the full power $P_{all}$, which was obtained by summing all powers from 4 to 50 Hz. We calculated the *RPL* as follows:(2)$$RPL_{i}=\frac{P_{i}}{P_{all}}$$

$P_{i}$ represented the energy of the *i*th frequency band, which was calculated by the Formula (1).

*PSE* reflects the distribution of the power spectrum and belongs to the information entropy of the frequency domain [25]. The higher *PSE* value indicates that the signal is more complex and disordered. *PSE* was calculated as follows:

Firstly, we calculated the frequency spectrum $X\left(\omega_{i}\right)$ of resting-state EEG signals by the periodogram method. $\omega_{i}$ represented the *i*th frequency band. Then, we calculated the power spectrum density:(3)$$PSD\left(\omega_{i}\right)=\frac{1}{N}{\left|X\left(\omega_{i}\right)\right|}^{2}$$

Normalized the power spectral density as shown in Formula (4):(4)$$PSD_{i}=\frac{PSD\left(\omega_{i}\right)}{\sum_{i}PSD\left(\omega_{i}\right)}$$

The *PSE* could be calculated by the standard entropy formula:(5)$$PSE_{i}=-\overset{n}{\underset{i=1}{\sum }}PSD_{i}\ln PSD_{i}$$

*LZC* measures the complexity of the signals by measuring the repeatability of the time series [26], which has been used to identify users’ emotional states [27]. The larger the *LZC* is, the faster the new patterns appear in the representation time series, which means the system is more complex. The calculation process of *LZC* was as follows. The EEG time series data were converted into a binary sequence C(n) and the initial value was defined as 1. The traversal process of intertemporal sequence EEG points could refer to [26]. Before data conversion, we used four Butterworth band-pass filters to obtain the theta, alpha, beta and gamma frequency bands’ EEG signals, respectively. Then we obtained the *LZC* of each frequency band:(6)LZC=C(n)nlog2n

As the nonlinear dynamics features, *PSE* and *LZC* reflect the complexities of the analyzed system from two different perspectives. A Pearson correlation analysis was used for the correlation analysis. We utilized the independent sample t-test for the significance analysis. Moreover, we used the Multi-SVM method to conduct the screening model of the MI performance.

## 3. Results

### 3.1. Correlation Analysis

In Figure 3, we found that all three features calculated in this paper (*RPL*, *PSE*, *LZC*) had the strongest and most significant correlation with left/right hand MI performance in the alpha band, whether for the eyes-open resting-state EEG or the eyes-closed state. The maximum correlations were r = 0.5 for *RPL*, r = −0.53 for PSE and r = −0.46 for *LZC* at the C4 channel. Pearson correlation coefficients between *RPL* and MI performance showed a significant positive correlation in most channels. The *PSE* and *LZC* of all channels showed a significant negative correlation with MI performance.

The results of the correlation between *RPL*, *PSE* and *LZC*, respectively, and both hands/feet MI performance were similar to the results of left/right hand MI performance (Figure 4). That was, all features in the alpha band showed significant correlation with MI performance. The maximum correlations were r = 0.36 of RPL at Cp4 channel, r = −0.49 of *PSE* at C4 channel, and r = −0.39 of *LZC* at the Cp4 channel in the alpha band. In short, *RPL* was positively correlated, while *PSE* and *LZC* were negatively correlated with BCI performance. The statistical significance levels by *p*-value in the alpha band were high enough to conclude. We could infer that the alpha bands may offer more information related to the user’s MI ability.

For the four classes of MI tasks, we performed the repeated measurement ANOVA among the absolute values of correlation coefficients in different frequency bands. We used SPSS for performing the repeated-measures ANOVA. Mauchly’s test of sphericity (*p* < 0.001) showed that the sphericity assumption was violated. After Greenhouse–Geisser correction, there was significant difference (*p* < 0.001) among the absolute values of the correlation coefficients in different frequency bands, whether for the two resting states (eyes open and eyes closed) or the three features (*RPL*, *PSE*, *LZC*). Then, we performed a paired *t*-test between pairwise frequency bands for the absolute values of the correlation coefficients. We found that all three features (*RPL*, *PSE*, *LZC*) in the alpha band had the strongest correlation with four classes of MI performance significantly (*p* < 0.001), which was consistent with the above results of the two classes of MI tasks. Hence, the following results for the four classes of MI tasks were *RPL*, *PSE* and *LZC* at alpha rhythm.

Figure 5 shows the absolute values of the correlation coefficients between the RPL, PSE and LZC, respectively, on the channels corresponding to the motor cortex (Fz, Fc3, Fc4, C3, C4, C5, C6, Cp3, Cp4, Cp5, Cp6, Cpz) and the four classes of MI performance. It was obvious that the correlations of the eyes-open resting-state EEG features were significantly stronger than the eyes-closed resting-state EEG (Greenhouse–Geisser-corrected repeated-measures ANOVA, *p* < 0.001). The two channels with the greatest correlation coefficients were C3 (r = 0.47) and C4 (r = 0.49) for *RPL*; C4 (r = −0.62) and Cp4 (r = −0.58) for *PSE*; and C3 (r = −0.45) and Cp4 (r = −0.47) for *LZC*. From the above results, it could be seen that the most relevant channels were mainly located in the channels corresponding to the sensorimotor cortex. For the two non-linear dynamic features, *PSE* and *LZC* were negatively correlated with MI performance in alpha rhythm, which indicated that the brain neural oscillations of people with weak MI ability were more disordered and complex in the resting state.

We calculated the average value of the two channels of each feature (C3 and C4 for *RPL*, C4 and Cp4 for *PSE*, C3 and Cp4 for *LZC*) as the feature value of each subject. Figure 6 shows the mean feature value and standard deviations of each group. The features of different subjects’ groups were compared and analyzed. The independent sample *t*-test was used for the statistical analysis. For *RPL*, *PSE* and *LZC* features, there were significant differences between group H and group M (*p* < 0.001) and group H and group L (*p* < 0.001), while group M and group L did not show statistical significance. These results showed that the high MI performance group and the low MI performance were distinguishable. We could also see that the feature distributions of different groups were not linear, which was the same as the previous study [17].

### 3.2. Screening Model of the MI Performance

We selected the alpha band eyes-open *RPL* on C3 and C4, *PSE* on C4 and Cp4, and *LZC* on C3 and Cp4 of each subject to generate a classifier to separate the high MI performance group (group H) from the low MI performance group (group L). The ten-fold classification accuracy of the combination of the above characteristics was 84.05%. We obtained the accuracy of 80.00% for *RPL*, 85.24% for *PSE* and 81.19% for *LZC*. The PSE performed the best among all characteristics for the screening of the MI performance.

We also constructed the three-class screening model of the MI performance using the above characteristics. Figure 7 shows the prediction distribution, which could reflect the inner-category classification results for the three MI performance groups. It shows that the misclassification which occurred among the three groups was imbalanced. For the three-class screening model, group H and group L were predictable. However, group M was easily predicted to be group L, which meant that the resting-state EEG patterns of group M were similar to group L. Hence, the distribution of EEG characteristics correlated to the MI performance was nonlinear, which was consistent with Figure 6.

## 4. Discussion

From the results of the correlation between *RPL*, *PSE*, *LZC* and the MI performance, we could see that for each characteristic the most significant correlation was in alpha rhythm. Previous studies have shown that MI is accompanied by changes in alpha and beta rhythm EEG in the motor-related cortex, which is always named mu or beta ERD. The amplitude of alpha (or mu) rhythm oscillations significantly decreased over the motor regions that began in the motor preparation stage [28], which indicated that alpha rhythm was more relevant to motor planning/programming. In addition, cross-frequency coupling played an important role in brain neural computation and communication, in which low-frequency neural oscillation signals acted as global interactions between brain regions form, and high-frequency neural oscillation signals were usually relatively localized activity [29]. It could be inferred that, compared with beta rhythm, alpha rhythm was more important for the trans-regional transmission and integration of motor information. Hence, alpha rhythm tended to recruit neurons in larger cortical areas under MI than beta rhythm.

Some studies found that the resting-state EEG alpha rhythm power of professional athletes was significantly higher than that of non-professional athletes or beginners [30], and it was related to motor control and final motor performance [31]. The analysis results in this paper also showed that the *RPL*, *PSE* and *LZC* in resting-state alpha rhythm were most relevant to MI performance. These results illustrated that alpha rhythm played an important role in motor-related mental information processing. In addition, we found that the correlations of eyes-open resting-state EEG features were stronger than eyes-closed resting-state EEG, which may be due to the interference of the increased occipital alpha-band (~10 Hz) power induced by the eyes closed state [32].

In this paper, we firstly used *LZC* as the neural maker to predict the MI performance and obtained an accuracy of 81.19%. The results demonstrated the feasibility of *LZC* for MI performance prediction. Ahn et al. utilized the *RPL* features of theta and alpha bands and obtained a classification accuracy of 82.35% for the separation of the high MI performance group from the low MI performance group among 34 subjects [17]. Theoretically, the power calculated in the time domain is the same as the power calculated in the frequency domain. We also calculated the *RPL* using *PSD*. The correlation results of the *RPL* calculated in the frequency domain were similar to those calculated in the time domain—that was, *RPL* had the strongest and most significant correlation with MI performance in the alpha band for the eyes-open resting-state EEG state. Due to the larger sample size, the screening model constructed in this paper had higher robustness. Moreover, we compared the features of upper and lower limbs MI tasks to reach a more convincing conclusion. The *PSE* result obtained in this paper was not consistent with the study of Zhang et al. [18]. In this paper, *LZC* was also the nonlinear dynamics, which confirmed the *PSE* result. In addition, the number of subjects was 40 in [18], which was less than half of our sample. From this point of view, our results were more reliable. Furthermore, previous research has shown that MI performance was correlated with subjects’ basic characteristics (gender, age, lifestyle, etc.) [33,34] and psychological states (motivation, self-confidence, frustration, etc.) [13,14]. This may be another reason for inconsistent results that need to be further explored.

## 5. Conclusions

In this paper, we investigated the correlation between resting-state EEG with upper and lower limbs MI performance. The above results showed that the EEG characteristics of the alpha rhythm of eyes-open resting state had the most significant correlation. An efficient screening model of the high MI performance group and the low MI performance group was constructed based on the three features, i.e., *RPL*, *PSE* and *LZC*. The above research findings can be used to further explore neurophysiological markers related to MI performance and to design tailored MI training strategies.

## Figures and Tables

**Figure 1 entropy-24-01556-f001:**
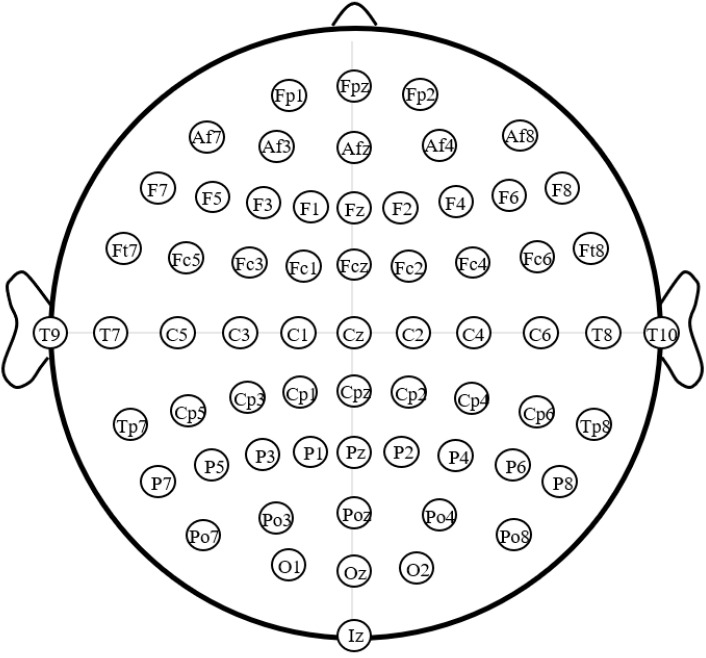
Position of 64-electrodes [21,22].

**Figure 2 entropy-24-01556-f002:**
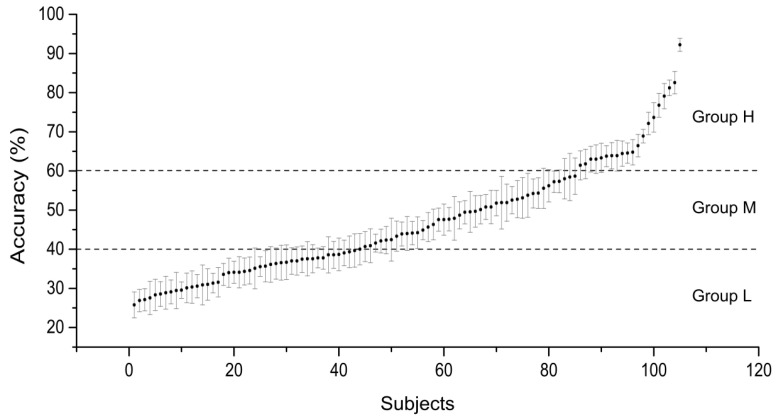
The classification accuracies of four MI tasks (left hand, right hand, both hands and feet). Subjects were assigned to three groups according to their classification accuracies.

**Figure 3 entropy-24-01556-f003:**
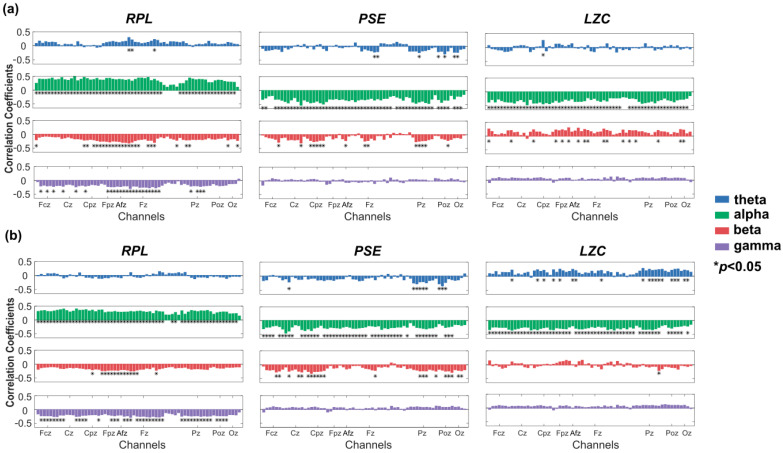
Pearson correlation coefficients between RPL, PSE and LZC, respectively, and left/right hand MI performance (* *p* < 0.05). (**a**) The correlation coefficients were calculated using eyes-open resting-state EEG. (**b**) The correlation coefficients were calculated using eyes-closed resting-state EEG.

**Figure 4 entropy-24-01556-f004:**
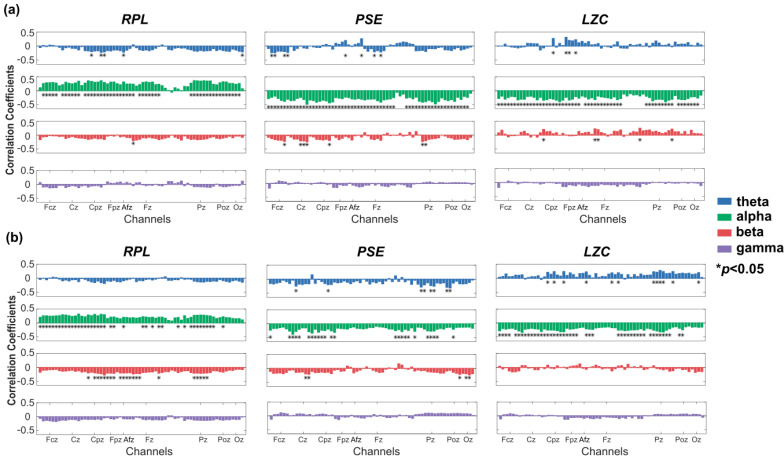
Pearson correlation coefficients between *RPL*, *PSE* and *LZC*, respectively, and both hands/feet MI performance (* *p* < 0.05). (**a**) The correlation coefficients were calculated using eyes-open resting-state EEG. (**b**) The correlation coefficients were calculated using eyes-closed resting-state EEG.

**Figure 5 entropy-24-01556-f005:**
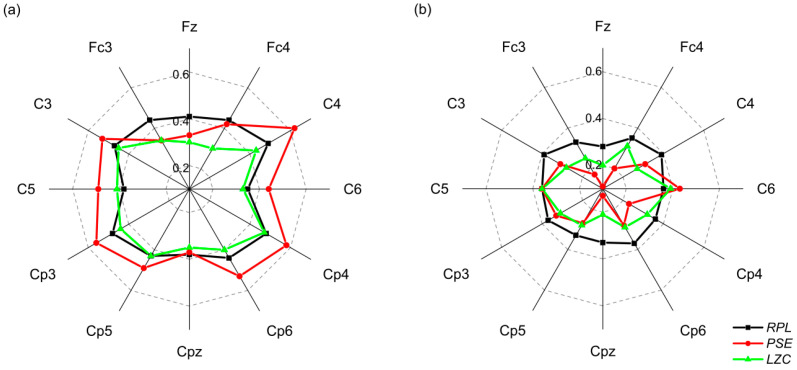
The absolute values of correlation coefficients between Fz, Fc3, Fc4, C3, C4, C5, C6, Cp3, Cp4, Cp5, Cp6, Cpz channels’ *RPL*, *PSE*, *LZC*, respectively, and four classes of MI performance. (**a**) The correlation coefficients were calculated using open-eyes resting-state EEG. (**b**) The correlation coefficients were calculated using closed-eyes resting-state EEG.

**Figure 6 entropy-24-01556-f006:**
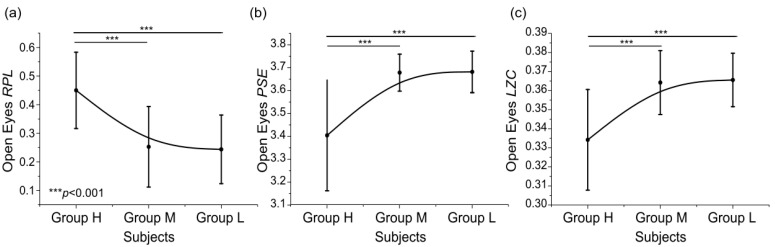
Distribution of *RPL* (**a**), *PSE* (**b**) and *LZC* (**c**) of alpha rhythm in eyes-open resting state among three groups (*** *p* < 0.001).

**Figure 7 entropy-24-01556-f007:**
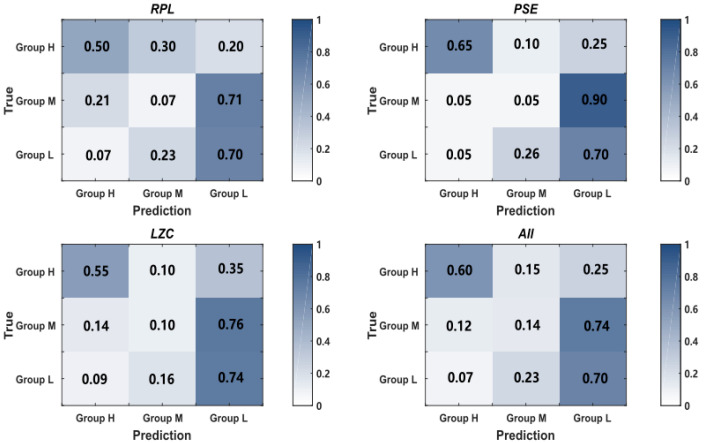
The prediction distribution of the three-class screening model of the MI performance using the *RPL*, *PSE*, *LZC* and all features.

## Data Availability

Publicly available datasets were analyzed in this study. These data can be found here: https://www.physionet.org/content/eegmmidb/1.0.0/; http://www.bci2000.org; http://circ.ahajournals.org/cgi/content/full/101/23/e215, accessed on 9 October 2019.

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
