# Peer review of "Resting-State EEG in Alpha Rhythm May Be Indicative of the Performance of Motor Imagery-Based Brain–Computer Interface"

_entropy, 2022, doi:10.3390/e24111556_

Round 1

Reviewer 1 Report

This paper investigates the problem of predicting the performance of motor imagery tasks with resting state EEG features. Although some results are not very strong, I think the overall story holds given the results presented, which indicates that alpha rhythm may be relevant to the mental information processing for the motor imagery tasks. Such results are enlightening for future studies and worth to pursue further.

Major

Please add a little more details about Multi-SVM for the screening model of the MI performance in Methods: type of kernel function, hyperparameter tuning, etc.

Figure 6: group A has the highest RPL, and the lowest PSE and LZC, any interpretation?

Section 3.2: the aim of this analysis is not to better discriminate subjects from different groups. In this sense, besides the SVM results (assuming you are using nonlinear kernel), showing the results from a linear classifier (LDA) would be helpful to understand the inherent properties of the data. Alternatively, you can also show the results using an LDA and a nonlinear neural network-based classifier.

Minor

Page 2, Line 93 and Page 3, Line 97 (caption of Figure 1): use “position of electrodes” instead of “electrode distributions”.

Page 3, Line 100: “we firstly filtered the EEG data of MI tasks from 8Hz to 30Hz”, I assume you mean filtering the EEG data with a band-pass filter (8 Hz ~ 30 Hz), please rephrase.

Page 3, Line 110: “middle” should be “medium”.

Page 7, Lines 196, 197: don’t use “based on”, rephrase.

Figure 6: error bars are standard deviations, correct?

Say “EEG features”, not “EEG characteristics”. Also call RPL, PSE, LZC etc. features instead of characteristics.

Author Response

Thanks so much for your time and your valuable suggestions. We really appreciate your patience to think so deeply about our work. Your advices are helpful for us to improve the manuscript. We have carefully revised the corresponding parts according to your suggestions. The detailed responses and corrections are as follow.

Major issues:

  1. Please add a little more details about Multi-SVM for the screening model of the MI performance in Methods: type of kernel function, hyperparameter tuning, etc.

We used LIBSVM to build the multiclass classifier. We selected linear kernel type and the default SVM type, i.e., C-SVC. We set the penalty factor C to 1. This setting is the commonly used and robust SVM type. We have made some modifications to make it more clearly (Line 106-108). “A linear support vector machine (SVM) was used to build the multiclass classifier with the help of the famous software package LIBSVM. We selected the default SVM type and set the penalty factor C to 1.”

  1. Figure 6: group A has the highest RPL, and the lowest PSE and LZC, any interpretation?

RPL reflected the magnitude of EEG power. PSE reflects the distribution of the power spectrum and belongs to the information entropy of the frequency domain. LZC measures the complexity of signals by measuring the repeatability of time series. Hence, group A (revision for group H now) has the highest RPL and the lowest PSE and LZC, which implies a concentrated distribution spectrum of EEG signal and less chaotic than other two groups. To be honest, we were confused with these results at beginning. Before the data analysis, we hoped the group A had the highest RPL, PSE and LZC at the same time. We asked experienced professors to check our data processing procedures and didn’t find any problems. So, we tried to make some interpretation in the discussion (Line 283-290). Nevertheless, the most important conclusion of this paper was that the alpha bands might offer information related to the user’s MI ability. The inherent causes of the change of the features among different groups need to be further researched by elaborate experimental paradigm and other signal acquisition and analysis methods.

  1. Section 3.2: the aim of this analysis is not to better discriminate subjects from different groups. In this sense, besides the SVM results (assuming you are using nonlinear kernel), showing the results from a linear classifier (LDA) would be helpful to understand the inherent properties of the data. Alternatively, you can also show the results using an LDA and a nonlinear neural network-based classifier.

Thank you for your advice. In this paper, we used linear kernel, as we answered in question 1. To further investigate the inherent properties of the data. We calculated the classification accuracies of our data using radial basis SVM (nonlinear) and LDA. We used the default parameters for the radial basis SVM to build a general and easy-to-use model. We obtained the accuracies of each feature and method as shown in the table. All three methods showed relatively similar results, although the classification accuracies of SVM were a little higher than that of LDA. Therefore, whether the classification model is linear or not does not affect the results.

Subject

Linear SVM

(This paper)

Radial basis SVM

LDA

RPL

80.00%

79.29%

77.78%

PSE

85.24%

85.95%

82.54%

LZC

81.19%

80.71%

80.95%

All features

84.05%

84.29%

82.54%

Minor issues:

  1. Page 2, Line 93 and Page 3, Line 97 (caption of Figure 1): use “position of electrodes” instead of “electrode distributions”.

Amended as suggested.

  1. Page 3, Line 100: “we firstly filtered the EEG data of MI tasks from 8Hz to 30Hz”, I assume you mean filtering the EEG data with a band-pass filter (8 Hz ~ 30 Hz), please rephrase.

We have checked the manuscript and revised the expression.

The EEG signals of MI tasks were first filtered by an 8-30 Hz Butterworth band-pass filter.

  1. Page 3, Line 110: “middle” should be “medium”.

Amended as suggested.

  1. Page 7, Lines 196, 197: don’t use “based on”, rephrase.

We have revised the similar expression in the manuscript.

(a) The correlation coefficients were calculated using open eyes resting-state EEG.

(b) The correlation coefficients were calculated using closed eyes resting-state EEG.

  1. Figure 6: error bars are standard deviations, correct?

In the figure 6, error bars are indeed standard deviations. We have added this information in the Results section.

  1. Say “EEG features”, not “EEG characteristics”. Also call RPL, PSE, LZC etc. features instead of characteristics.

I have checked the full text and revised the expression according to your suggestions.

Reviewer 2 Report

Authors showed correlation between resting-state EEG with upper and lower limbs motor imagery performance. The simulated results with analysis looks appropriate. English grammar looks fine. However, authors need to update some answers for the following comments before publication. Thus, the manuscript could be minor revision.

1. Abstract format need to be corrected. Please check MDPI format.

2. For Figure 6, label fonts are too small to be seen.

3. Author contribution and data availability sections are missing.

4. Authors must use abbreviated journal names in the reference section.

5. Figures 3 and 4 fonts need to be increased for better clarity.

6. I am wondering why authors select 64-electrode in Figure 1.

7. Why authors select PRL, PSE, LZC parameters in Pearson correlation coefficients ? Are there any remaining parameters ?

Author Response

Thanks a lot for reviewing our manuscript. Your valuable advices are helpful for us to improve the manuscript. We have revised the manuscript according to your comments. We hope the revised manuscript is satisfying. The answers to your comments are listed point by point as follow.

  1. Abstract format need to be corrected. Please check MDPI format.

Thank you for your reminder. We have revised the abstract according to the MDPI format.

Abstract: Motor imagery-based brain-computer interfaces (MI-BCIs) have great application pro-spects in motor enhancement and rehabilitation. However, the capacity to control a MI-BCI varies among persons. Predicting the MI ability of a user remains challenging in BCI studies. We first calculated the relative power level (RPL), power spectral entropy (PSE) and Lempel-Ziv com-plexity (LZC) of resting-state open and closed-eye EEG of different frequency bands and inves-tigated their correlations with the upper and lower limbs MI performance (left hand, right hand, both hands and feet MI tasks) on as many as 105 subjects. Then, the most significant related features were used to construct a classifier to separate the high MI performance group from the low MI performance group. The results showed that the features of open-eye resting alpha-band EEG had the strongest significant correlations with MI performance. The PSE performed the best among all features for the screening of the MI performance, with the classification accuracy of 85.24%. These findings demonstrated the alpha bands might offer information related to the user’s MI ability, which could be used to explore more effective and general neural markers to screen subjects and design individual MI training strategies.

  1. For Figure 6, label fonts are too small to be seen.

Thank you for your advice. We further increased the fonts of the labels as follow.

  1. Author contribution and data availability sections are missing.

Thank you for your reminder. I have added the author contribution and data availability sections at the end of the paper.

Author Contributions: Conceptualization, Kun Wang, Minpeng Xu and Dong Ming; Data curation, Shanshan Zhang; Formal analysis, Kun Wang and Feifan Tian; Funding acquisition, Minpeng Xu and Dong Ming; Investigation, Kun Wang; Methodology, Kun Wang and Shanshan Zhang; Re-sources, Lichao Xu; Supervision, Minpeng Xu; Writing – original draft, Feifan Tian; Writing – review & editing, Kun Wang.

Data Availability Statement: Publicly available datasets were analyzed in this study. This data can be found here: https://www.physionet.org/pn4/eegmmidb/; http://www.bci2000.org; http://circ.ahajournals.org/cgi/content/full/101/23/e215.

  1. Authors must use abbreviated journal names in the reference section.

Amended as suggested.

  1. Figures 3 and 4 fonts need to be increased for better clarity.

Thank you for your advice. We further increased the fonts of the labels as follow. On the premise that the electrode names do not cover each other, the font has been adjusted to the maximum.

  1. I am wondering why authors select 64-electrode in Figure 1.

All EEG data used in this paper were from EEG Motor Movement/Imagery Dataset. Subjects performed different motor/imagery tasks while 64-channel EEG were recorded using the BCI2000 system (Schalk, G., McFarland, D.J., Hinterberger, T., Birbaumer, N., Wolpaw, J.R. BCI2000: A General-Purpose Brain-Computer Interface (BCI) System. [http://www.ncbi.nlm.nih.gov/pubmed/15188875] IEEE Transactions on Biomedical Engineering 51(6):1034-1043, 2004). The selection and distribution of electrodes have been determined in this public database. We have added references in the title of Figure 1.

  1. Why authors select PRL, PSE, LZC parameters in Pearson correlation coefficients ? Are there any remaining parameters ?

In the previously studies, RPL and PSE have been selected to investigate the correlations of MI performance (Ahn M; Cho H. High theta and low alpha powers may be indicative of BCI-illiteracy in motor imagery. PloS one. 2013,8(11),e80886. Zhang R; Xu P. Predicting inter-session performance of SMR-based brain–computer interface using the spectral entropy of resting-state EEG. Brain Topogr. ,2015,28(5),680-90). However, these studies are based on the left and right hand MI tasks. For the practical application of MI-BCI, the recognition of lower limbs is as important as that of upper limbs. Hence, we first selected RPL and PSE in our study to verify whether the correlation conclusions under both upper and lower limbs MI tasks are consistent with previous studies. LZC is often used as an EEG feature to detect diseases such as depression. Therefore, to explore the new resting-state EEG feature related to the user's motor imagery performance, we selected LZC which is a non-linear dynamic feature like PSE. We have made a detailed explanation in the introduction.

Our results demonstrated the alpha bands might offer information related to the user’s MI ability, which could be used to explore more effective and general neural markers. Hence, there may be other parameters associated with MI abilities. We will further explore new features in the EEG data of alpha band in the future.

Reviewer 3 Report

The authors of "Resting-state EEG in alpha rhythm may be indicative of the performance of motor imagery-based brain-computer interface" aim to create a predictor of BCI-literacy using resting state data. Their results show that they are able to distinguish high performers from medium and low performers, preferably using the resting state with eyes open data. The manuscript seems interesting, but I have several methodological issues and questions, and a couple of comments on the results.

Methodological issues:
* For the MI tasks, the participants had their eyes open or closed?
* Data is said to be acquired at a sampling rate of 160 Hz and a 80-Hz low-pass filter is said to be applied at preprocessing. I hope that is not the case, as the 80 Hz filter should be applied before sampling. Plus, the use of a 80-Hz filter for a 160-Hz sampling rate is too optimistic, as filters are not ideal. Please, define the *exact* configuration of the acquisition.
* There is no description about the reference, either online nor (if re-referenced was performed) offline. Please, add this information.
* I do not understand line 121: "For the EEG data of each channel, every 1500 points were epoched as a sample."
* No information is provided about the power line frequency or how it was dealt with. Was it 50 or 60 Hz? Was it removed by using a notch filter? The power line is the largest interference in a normal EEG, and should be corrected.
* The description of the calculation of the power spectrum (lines 135 to 140) should appear before the description of the relative power, as it uses the power spectrum as starting point. Furthermore, Pi in Eq. 3 is the same as RPLi in Eq. 2, and completely different from Pi in Eq. 2. This must be corrected.
* Lines 147 to 148 read: ". The traversal process of intertemporal sequence EEG points was used to update the ?(?)". This sentence gives no information at all. If the authors do not want to explain the (quite convoluted) calculation of LZC they should cite a paper describing it, but including such a sentence is more confusing than clarifying.
* I suppose the filtering described in lines 119 and 120 apply only to the LZC, as the spectral measures must be calculated over the broadband, untouched data. This should be clarified in the manuscript. It also should be indicated how the relative power per band was calculated. The descriptions given in this section are very poor.
* The authors do not indicate any multiple comparison correction. This correction is mandatory, as they calculate parameters for 64 channels, 4 bands, and 3 metrics in 4 different MI tasks and 2 different resting state tasks.

Questions about the results:
* In the paragraph starting in line 158, all results are said to be described for alpha band. But in the paragraph starting in line 169 is is not so clear.
* I would re-label the A, B, and C groups to represent their BCI literacy or the accuracy of the classifier, using for example H, M, and L for high, medium, and low accuracy, respectively. It would be easier to understand the graphs and confusion matrices.

And a couple of general comments:
* The correlation between performance and relative power is larger a) in motor areas and b) in the alpha band. Being motor areas, I would relate this with the mu rhythm, not the alpha one. This is supported by the finding that eyes open (with lower occipital alpha) are stronger than with eyes closed (with alpha power interfering in the measurement of the mu rhythm).
* My second commentary is a question or reflection. The authors say that training can improve the BCI literacy of the participants. Does the database that the authors have used include post-training recordings? Do they think that the training will change the intensity of the alpha/mu rhythm, working as a neurofeedback training?

Last, the English language is quite good, but I have found some strange expressions. I recommend the authors to have the manuscript proofchecked by a native English speaker. In addition, I found some minor typos:
* In line 58 there is no point after "et al.".
* In line 81 the list of task should be "(left hand, right hand, both hands and both feet)".
* In the database description, it is weird for me talking about resting state sessions and task sessions, as all the measurements are taken in the same session. It would be better to use other words, as "recording" or "scan", instead of "session".
* You do not need to use the phrasing "in this study" in the methods sections, as this section describes the methods used in the current study.
* In line 162 there is a huge spacing between "C4" and "channel".

Author Response

Thanks a lot for reviewing my manuscript in your busy schedule. Your valuable advices are helpful for us to improve the manuscript. We have tried to make some improvements according to your comments. We hope the new submission is satisfying. The answers to your comments are listed point by point as follows.

Methodological issues:

  1. For the MI tasks, the participants had their eyes open or closed?

All EEG data used in this paper were from an open dataset named EEG Motor Movement/Imagery Dataset (https://www.physionet.org/pn4/eegmmidb/). There is no description about whether the subjects open or close their eyes in the MI task in the instruction document. We also checked other literatures using this public dataset, and didn’t find relevant description. The 10Hz rhythmic EEG signal caused by eye closed could affect the MI induced EEG pattern. Hence, in MI experiment, subjects are usually required to open their eyes.

  1. Data is said to be acquired at a sampling rate of 160 Hz and a 80-Hz low-pass filter is said to be applied at preprocessing. I hope that is not the case, as the 80 Hz filter should be applied before sampling. Plus, the use of a 80-Hz filter for a 160-Hz sampling rate is too optimistic, as filters are not ideal. Please, define the *exact* configuration of the acquisition.

Thank you very much for your questions. To be honest, we made a mistake in the writhing of the paper. When we discussed the preprocessing method before, a colleague may say that we can use the 80Hz low-pass filter. But in our actual processing, we did not use the low-pass filter. We have carefully checked the program and did not use any low-pass filtering in the pre-processing. We have corrected the errors in the manuscript.

  1. There is no description about the reference, either online nor (if re-referenced was performed) offline. Please, add this information.

All data were collected under offline experimental conditions. we have added relevant explanations in the Database introduction part.

  1. I do not understand line 121: "For the EEG data of each channel, every 1500 points were epoched as a sample."

The sampling rate was 160Hz. Hence, for the one-minute resting task, we could record about 9600 data points of each channel. We divided the recorded data into 6 samples, and every 1500 points were taken as a sample according to the convention. We have revised this expression in the paper.

  1. No information is provided about the power line frequency or how it was dealt with. Was it 50 or 60 Hz? Was it removed by using a notch filter? The power line is the largest interference in a normal EEG, and should be corrected.

Indeed, a 50 Hz or 60 Hz notch filter are usually used to remove power frequency interference when collecting data. However, we did not find the relevant signal processing steps in the description document of this dataset. In fact, we randomly selected some samples to plot the spectrum diagram. The results showed no power frequency interference components existed. Besides, the highest frequency band range in this study, i.e., gamma band, is 30~50 Hz. After band-pass filtering, the data could hardly be affected by 50Hz components. Hence, we didn’t use notch filter.

  1. The description of the calculation of the power spectrum (lines 135 to 140) should appear before the description of the relative power, as it uses the power spectrum as starting point. Furthermore, Pi in Eq. 3 is the same as RPLi in Eq. 2, and completely different from Pi in Eq. 2. This must be corrected.

For your first question, we directly used time-domain EEG data after band-pass filtering to obtain power feature. We first used four Butterworth band-pass filters to obtained the theta, alpha, beta, and gamma bands EEG signals respectively. Then, we calculated the power of each frequency band as follow:

                                   (1)

We normalized the band powers by using the full power  , which was obtained by summing all powers from 4~50 Hz. We calculated the RPL as follow:

                                    (2)

 represented the energy of the ith frequency band which was calculated by the formula (1).

For your second question, we have revised the symbols in the formula to avoid misunderstanding. We used  to express the Normalized power spectral density and  to express the power spectrum density of ith frequency band.

We have revised the above contents in the text.

  1. Lines 147 to 148 read: ". The traversal process of intertemporal sequence EEG points was used to update the ?(?)". This sentence gives no information at all. If the authors do not want to explain the (quite convoluted) calculation of LZC they should cite a paper describing it, but including such a sentence is more confusing than clarifying.

Thank you for your suggestion. The calculation process of LZC is quite convoluted. We have cited a paper in the manuscript. The traversal process of intertemporal sequence EEG points could refer to [26].

[26] Aboy M; Hornero R. Interpretation of the Lempel-Ziv complexity measure in the context of biomedical signal analysis. IEEE Trans. Biomed. Eng., 2006,53(11),2282-8

  1. I suppose the filtering described in lines 119 and 120 apply only to the LZC, as the spectral measures must be calculated over the broadband, untouched data. This should be clarified in the manuscript. It also should be indicated how the relative power per band was calculated. The descriptions given in this section are very poor.

Butterworth band-pass filters were applied to both the RPL and LZC. For the PSE, periodogram method was used to calculate power spectrum. The relative power per band was calculated refer to answer of question 6. And we have carefully improved the descriptions of this section. We hope you are satisfied.

  1. The authors do not indicate any multiple comparison correction. This correction is mandatory, as they calculate parameters for 64 channels, 4 bands, and 3 metrics in 4 different MI tasks and 2 different resting state tasks.

Thank you for your advice. we have given a detailed explanation in the results section (Line 193-204).

We have made the repeated measurement ANOVA among the absolute values of correlation coefficients in different frequency bands. We used SPSS for making repeated-measures ANOVA. Mauchly's test of sphericity (p<0.001) showed the sphericity assumption has been violated. After Greenhouse-Geisser corrected, there was significant difference (p<0.001) among the absolute values of correlation coefficients in different frequency bands no matter for the two-resting state (eyes open and eyes closed) or the three features (RPL, PSE, LZC). Then we performed paired t-test between pairwise frequency bands for absolute values of correlation coefficients. We found that all three features (RPL, PSE, LZC) in the alpha band had the strongest correlation with four classes MI performance significantly (p<0.001), which was consistent with the above results of two classes MI tasks. Hence, the main results for the four classes MI tasks were RPL, PSE and LZC at alpha rhythm.

We also made the repeated measurement ANOVA between the absolute values of correlation coefficients for the two different resting state tasks (eyes open and eyes closed). Mauchly's test of sphericity (p<0.001) showed the sphericity assumption has been violated. After Greenhouse-Geisser corrected, there was significant difference (p<0.001) between the absolute values of correlation coefficients for the two different resting state tasks. The correlations of eyes open resting-state EEG features were significantly stronger than eyes closed resting-state EEG.

Questions about the results:

  1. In the paragraph starting in line 158, all results are said to be described for alpha band. But in the paragraph starting in line 169 is not so clear.

Thank you for reminding. We have made the further elaboration about the results of the paragraph starting in line 178. The results of the correlation between RPL, PSE, and LZC, respectively, and both hands/feet MI performance were similar to the results of left/right hand MI performance (figure 4). That was all features in alpha band showed significant correlation with MI performance. The maximum correlations were r=0.36 of RPL at Cp4 channel, r=-0.49 of PSE at C4 channel, and r=-0.39 of LZC at the Cp4 channel in alpha band.

  1. I would re-label the A, B, and C groups to represent their BCI literacy or the accuracy of the classifier, using for example H, M, and L for high, medium, and low accuracy, respectively. It would be easier to understand the graphs and confusion matrices.

Thank for your suggestion. We have labeled the H, M and L groups for high, medium, and low accuracy, respectively in the text and figure 2,6, and 7.

Figure 2. The classification accuracies of four MI tasks (left hand, right hand, both hands and feet). Subjects were assigned to three groups according to their classification accuracies

Figure 6. Distribution of RPL, PSE and LZC of alpha rhythm in eyes open resting state among three groups (***p<0.001).

Figure 7. The prediction distribution of the three-class screening model of the MI performance using the RPL, PSE, LZC and all features.

And a couple of general comments:

  1. The correlation between performance and relative power is larger a) in motor areas and b) in the alpha band. Being motor areas, I would relate this with the mu rhythm, not the alpha one. This is supported by the finding that eyes open (with lower occipital alpha) are stronger than with eyes closed (with alpha power interfering in the measurement of the mu rhythm).

Basically, motor imagery could activate neural networks in the primary sensorimotor cortices which is manifested as blocking or desynchronization of 8–13 Hz and/or 18–26 Hz rhythm, always named mu or beta ERD (Pfurtscheller, G., Brunner, C., Schlögl, A., & Da Silva, F. L. (2006). Mu rhythm (de) synchronization and EEG single-trial classification of different motor imagery tasks. NeuroImage, 31(1), 153-159). Hence, mu rhythm usually refers to signals in the alpha band of the primary motor cortex induced by motor imagery tasks. At the same time, authors have also used alpha ERD in many literatures which has the same meaning as mu ERD (Yuan, H., Liu, T., Szarkowski, R., Rios, C., Ashe, J., & He, B. (2010). Negative covariation between task-related responses in alpha/beta-band activity and BOLD in human sensorimotor cortex: an EEG and fMRI study of motor imagery and movements. Neuroimage, 49(3), 2596-2606; ZapaÅ‚a, D., Iwanowicz, P., Francuz, P., & Augustynowicz, P. (2021). Handedness effects on motor imagery during kinesthetic and visual-motor conditions. Scientific reports, 11(1), 1-12). We also made some revisions to indicate mu rhythm in introduction and discussion parts.

Thank you for your second comment. The power of occipital alpha-band (~ 10 Hz) brain waves is increased when peoples’ eyes are closed, rather than open (Hohaia, W., Saurels, B. W., Johnston, A., Yarrow, K., & Arnold, D. H. (2022). Occipital alpha-band brain waves when the eyes are closed are shaped by ongoing visual processes. Scientific Reports, 12(1), 1-10). Hence, the correlations of eyes open resting-state EEG features were stronger than eyes closed resting-state EEG, which may be due to the interference of the increased occipital alpha-band (~ 10 Hz) power induced by eyes closed. We have added this description in the discussion.

  1. My second commentary is a question or reflection. The authors say that training can improve the BCI literacy of the participants. Does the database that the authors have used include post-training recordings? Do they think that the training will change the intensity of the alpha/mu rhythm, working as a neurofeedback training?

Thank you for your comment. The research team from the Graz University of Technology analyzed near-infrared and EEG signals, and found that training with an MI-based BCI affects cortical activation patterns especially in users with low BCI performance (Kaiser V; Bauernfeind G. Cortical effects of user training in a motor imagery based brain-computer interface measured by fNIRS and EEG. Neuroimage, 2014,85(1),432-44). We also cited this literature in the introduction. Unfortunately, there is no post-training recordings in the public database we have used. Therefore, we didn’t know how subjects think about the working of neurofeedback training for the change of the intensity of the alpha/mu rhythm. Nevertheless, our recent experiment of MI-BCI based on virtual games showed that neurofeedback training could effectively improve the intensity of the alpha/mu rhythm (unpublished).

Last, the English language is quite good, but I have found some strange expressions. I recommend the authors to have the manuscript proofchecked by a native English speaker. In addition, I found some minor typos:

We have corrected the errors according to your advices and carefully improved the writing. We hope the revised manuscript is satisfying.

  1. In line 58 there is no point after "et al.".

Amended as suggested.

  1. In line 81 the list of task should be "(left hand, right hand, both hands and both feet)".

Amended as suggested.

  1. In the database description, it is weird for me talking about resting state sessions and task sessions, as all the measurements are taken in the same session. It would be better to use other words, as "recording" or "scan", instead of "session".

Thank you for your advice. All EEG data used in this paper were from EEG Motor Movement/Imagery Dataset which was created and contributed to PhysioNet by the developers of the BCI2000 instrumentation system. We carefully read the description documents of the dataset and used “run” instead of “session” in this paper (https://www.physionet.org/pn4/eegmmidb/).

  1. You do not need to use the phrasing "in this study" in the methods sections, as this section describes the methods used in the current study.

We have deleted the phrasing “in this study” in the methods sections.

  1. In line 162 there is a huge spacing between "C4" and "channel".

Amended as suggested.

Round 2

Reviewer 1 Report

I appreciate the authors’ efforts to improve the manuscript. The new version has resolved all of my concerns. 

Author Response

Thanks again for your valuable suggestions for improving the manuscript.  

Reviewer 3 Report

This is a re-review for "Resting-state EEG in alpha rhythm may be indicative of the performance of motor imagery-based brain-computer interface", where the authors aim to create a predictor of BCI-literacy using resting state data.

My first commentary is regarding the database information. I understand that the EEG acquisition was not made by the authors, but they need to know some information about the procedure.
Firstly, they are unsure regarding the state of the eyes during the task, although they suppose the IM task was recorded with eyes open (which makes sense, as they mention). This is an important piece of information, and they should ask the authors of the database if they are not sure.
Secondly, the authors say that the EEG was recorded "under offline experimental conditions". I have no idea what that means, but again, the reference used at recording time, and if this reference as modified during the processing of the data, is an important piece of information.
The authors must find out this information before the paper can be accepted. It is extremely important to know the data you are using, and the authors are lacking in this aspect.

Regarding the per-band calculation, calculating the per-band relative power using the time series is a very convoluted way, and prone to, when the computation is simple using the power spectrum. It is not inherently incorrect, but it is indeed a weird approach. And the authors use the PSD to calculate the entropy, so this approach is even more strange.
I firmly encourage the authors to re-compute the per-band power in the classical way, this is, taking the PSD (normalized between 4 and 30 Hz), and then accounting for the total (relative) power between 4 and 8 Hz (theta), between 8 and 13 Hz (alpha), between 13 and 30 Hz (beta) and between 30 and 50 Hz (gamma).

Last, regarding the 1500 points "taken as a sample", the authors should use the work "epoch", not "sample". A sample is a data point.

Author Response

Thanks a lot for reviewing my manuscript despite your busy schedule. Your valuable advice is helpful for us to improve the manuscript. We have tried to make some improvements according to your comments. We hope the new submission is satisfying. The answers to your comments are listed point by point in the attachment.

Round 3

Reviewer 3 Report

I would like to thank the authors for addressing all my commentaries.

Regarding the EEG acquisition, it is indeed unfortunate that some the information is missing. But I understand it is not the authors fault.

Regarding the spectrum analysis, I strongly recommend the authors, in the future, to use open software to avoid licensing issues. An expired license for the images of a manuscript in the publications process, while understandable taking into account the economy, is not acceptable. However, as the results of the new analysis seem to be congruent with the original ones, I think that no harm is done.

Last, I want to congratulate the authors on their work.